# Enhanced Aerosol Containment Performance of a Negative Pressure Hood with an Aerodynamic Cap Design: Multi-Method Validation Using CFD, PAO Particles, and Microbial Testing

**DOI:** 10.3390/bioengineering12060624

**Published:** 2025-06-09

**Authors:** Seungcheol Ko, Kisub Sung, Min Jae Oh, Yoonjic Kim, Min Ji Kim, Jung Woo Lee, Yoo Seok Park, Yong Hyun Kim, Ju Young Hong, Joon Sang Lee

**Affiliations:** 1School of Mechanical Engineering, Yonsei University, Seoul 03722, Republic of Korea; kscv0412@gmail.com; 2SS-ENG Co., Ltd., Bucheon 14449, Republic of Korea; kssung@ss-eng.net (K.S.); corono@naver.com (J.W.L.); 3AI & Energy Research Center, Korea Photonics Technology Institute, Gwangju 61007, Republic of Korea; hso5922@kopti.re.kr (M.J.O.); yonghyun@kopti.re.kr (Y.H.K.); 4Department of Emergency Medicine, Seoul National University College of Medicine, Seoul 03080, Republic of Korea; ambition86.kim@gmail.com; 5Emergency Medicine, Yonsei University College of Medicine, Seoul 03722, Republic of Korea; minzhi@yonsei.ac.kr (M.J.K.); pys0905@yuhs.ac (Y.S.P.)

**Keywords:** respiratory infection, computational fluid dynamics, intubation, aerosol transmission

## Abstract

Healthcare providers performing aerosol-generating procedures (AGPs) face significant infection risks, emphasizing the critical need for effective aerosol containment systems. In this study, we developed and validated a negative pressure chamber enhanced with an innovative aerodynamic cap structure designed to optimize aerosol containment. Initially, computational fluid dynamics (CFD) simulations were performed to evaluate multiple structural improvement ideas, including air curtains, bidirectional suction, and aerodynamic cap structures. Among these, the aerodynamic cap was selected due to its superior predicted containment performance, practical feasibility, and cost-effectiveness. The CFD analyses employed realistic transient boundary conditions, precise turbulence modeling using the shear stress transport (SST) k–ω model, and detailed droplet evaporation dynamics under realistic humidity conditions. A full-scale prototype incorporating the selected aerodynamic cap was fabricated and evaluated using physical polyalphaolefin (PAO) particle leakage tests and biological aerosol validation with aerosolized *Bacillus subtilis*. For the physical leakage tests, the chamber opening was divided into nine sections, and the aerosol dispersion was tested in three distinct directions: ceiling-directed, toward the suction hole, and opposite the suction hole. These tests demonstrated significantly stabilized airflow and substantial reductions in aerosol leakage, consistently maintaining containment levels below the critical threshold of 0.3%, especially under transient coughing conditions. The biological aerosol experiments, conducted in a simulated emergency department environment, involved aerosolizing bacteria continuously for one hour. The results confirmed the effectiveness of the aerodynamic cap structure in achieving at least a one millionth (10^−6^) reduction in the aerosolized bacterial leakage compared to the control conditions. These findings highlight the importance and effectiveness of advanced CFD modeling methodologies in accurately predicting aerosol dispersion and improving containment strategies. Although further studies assessing the structural durability, long-term operational ease, and effectiveness against pathogenic microorganisms are required, the aerodynamic cap structure presents a promising, clinically practical infection control solution for widespread implementation during aerosol-generating medical procedures.

## 1. Introduction

Healthcare providers in critical care and emergency medicine settings face significant risks of infection during pandemics, including the coronavirus disease 2019 (COVID-19) pandemic [1,2]. They frequently conduct aerosol-generating procedures (AGPs), including intubation, extubation, cardiopulmonary resuscitation (CPR), and airway suctioning. These procedures significantly increase their risk of infection owing to the high levels of aerosol exposure [2,3]. Despite implementing numerous protective measures, ensuring the safety of healthcare providers during AGPs remains a challenge in clinical practice. Various protective strategies and devices, including barrier enclosures and negative pressure hoods, have been developed and implemented to mitigate the aerosol transmission risks during AGPs [4,5]. These innovations show potential in reducing aerosol transmission; nevertheless, practical limitations remain regarding their effectiveness, comfort, ease of use, and applicability across different clinical settings [4,5,6,7]. Recent studies of barrier devices, such as aerosol boxes, have highlighted concerns, including restricted workspace, poor ergonomics, and impaired visualization, which can compromise safety and efficacy during aerosol-generating procedures (AGPs) [8,9,10]. Additionally, significant aerosol dispersion and subsequent exposure risks during the removal (“doffing”) of these devices have been reported, raising concerns regarding their practical effectiveness and potential secondary contamination [10,11]. Furthermore, recent evaluations have underscored the necessity of structural and functional improvements to balance protective performance with practical usability, thereby ensuring reliable clinical acceptance across diverse clinical scenarios [8,9,10].

Recent advances in computational fluid dynamics (CFD) modeling have emphasized the necessity of accurately modeling complex interactions between turbulent airflow and aerosol dispersion, realistic transient boundary conditions such as coughing behaviors, and detailed droplet evaporation dynamics under inhomogeneous humidity conditions [12,13,14]. These advanced modeling approaches offer critical insights into aerosol containment mechanisms, significantly addressing the limitations of traditional modeling methods. Specifically, sophisticated multi-component Eulerian–Lagrangian approaches have revealed the significant impact of detailed transient boundary conditions and complex turbulence modeling on the reliability of containment predictions. Furthermore, advanced ventilation strategies, such as stratum ventilation, have demonstrated superior efficacy in aerosol containment, suggesting potential improvements to traditional negative pressure isolation methods [15]. Recent investigations have also identified critical limitations of conventional negative pressure hoods, including inadequate modeling of human respiratory behaviors and environmental sensitivities, highlighting the need for innovative CFD-based solutions tailored for clinical scenarios [16,17,18]. These findings emphasize the urgent requirement for integrating more sophisticated modeling approaches and targeted containment strategies to overcome existing limitations and ensure the safety of healthcare providers.

Furthermore, traditional negative pressure isolation rooms require considerable resources for installation and maintenance. These rooms also have insufficient capacity during periods of increased patient volume, including pandemics or sudden surges in emergencies, highlighting the urgent need for more feasible and effective containment strategies. In our previous studies, we developed a negative pressure respiratory barrier enclosure specifically designed to enhance aerosol containment during high-risk medical procedures [19,20]. Using CFD simulations and particle image velocimetry (PIV) analysis, the airflow dynamics of the enclosure were optimized, resulting in a significant reduction in the aerosol leakage [20,21]. Physical leakage tests were conducted to confirm the consistent containment efficiency, showing particle leakage rates typically below 0.3%. This value corresponds to an aerosol containment efficiency of approximately 99% [20]. Infectious disease specialists confirmed that the level of aerosol containment provided by the enclosure exceeds the typical protection offered by standard personal protective equipment (PPE), including N95 respirators (approximately 95% efficiency). Therefore, the enclosure represents a potential complementary or partial replacement option in relation to standard PPE, providing additional safety for healthcare providers.

Additionally, previous CFD simulation results, although not perfectly identical numerically to experimental polyalphaolefin (PAO) leakage data, demonstrated strong qualitative agreement and consistent trends in the aerosol leakage patterns (Appendix A [20]). This consistency in the trends validated the CFD model’s ability to accurately predict the relative changes in the aerosol containment performance according to various structural modifications.

Despite these advancements, limitations have been observed in scenarios involving rapid and significant airflow changes, particularly during coughing compared to regular breathing. Although the average leakage remained consistently below 0.3%, the particle leakage during coughing simulations increased significantly, peaking at approximately 0.57%. This is more than twice the leakage observed under normal breathing conditions [20]. These results raised important concerns about the robustness of the containment under realistic clinical conditions involving dynamic airflow changes.

To address these concerns, three structural enhancements were selected following internal discussions: installing an air curtain, implementing bilateral suction, and adding an aerodynamic cap structure at the chamber opening. Additional CFD simulations were conducted to compare these proposed designs. Given the strong qualitative agreement between the CFD predictions and the experimental leakage results observed in previous studies, repeated simulations identical to the full-duration leakage tests were considered unnecessary. Instead, quick CFD simulations focusing on aerosol leakage for the initial 8 s were conducted to compare the relative containment performance trends among these structural options. The results demonstrated comparable and significant containment improvements for all three approaches. Considering the ease of implementation, cost-effectiveness, and practicality for immediate clinical use, the aerodynamic cap structure was ultimately selected as the optimal solution. Consequently, the present study aimed to validate the effectiveness of the newly developed aerodynamic-cap-enhanced negative pressure hood using rigorous physical and biological testing, providing a feasible and reliable approach for enhancing healthcare workers’ safety during AGPs.

## 2. Materials and Methods

### 2.1. Study Design and Overview

The overall validation flow of this study comprised five systematic steps, as depicted in Figure 1. Through initial brainstorming, the design team identified several candidate concepts—such as a front air curtain, a bidirectional suction system, and an aerodynamic cap—each hypothesized to enhance aerosol containment. Subsequently, computational fluid dynamics (CFD) simulations were performed to quantify and predict the aerosol leakage for each candidate model. The candidate designs meeting the predefined performance threshold were then ranked based on their manufacturability and cost-effectiveness. The highest-ranked design was mocked-up into a full-scale prototype, which was subsequently subjected to polyalphaolefin (PAO) particle leakage tests and biological aerosol assessments. This structured validation approach effectively integrated theoretical modeling with rigorous physical and biological experimentation, ensuring comprehensive evaluation of the aerosol containment performance.

### 2.2. Negative Pressure Hood Design

The negative pressure hoods utilized in this study were developed following a structured design and validation process based on our previously reported improved respiratory barrier enclosure [20], initially modified from an earlier design [19] to enhance the usability and aerosol containment during aerosol-generating procedures (AGPs) (Figure 2A).

First, preliminary evaluations, including computational fluid dynamics (CFD) simulations and smoke visualization tests, identified significant aerosol leakage, predominantly from the superior portion of the frontal opening. Based on these findings, an aerodynamic cap structure was designed according to the airflow dynamics and modeled using Autodesk AutoCAD LT 2020 (Autodesk, Inc., San Francisco, CA, USA). The aerodynamic cap specifically addressed the identified leakage points; it comprised a 90° sector shape with an approximate radius of 15 cm and an inward-curved quarter circle at the base (radius ≈ 2 cm). This secondary curve effectively redirected the boundary layer streamlines toward the suction manifold, significantly enhancing the aerosol particle recapture. For practical use, the cap was equipped with a slot-in engagement mechanism enabling rapid installation and removal from the previously validated negative pressure chamber (Figure 2B).

Negative pressure within the hood was continuously generated using an 80 mm duct connected to a negative pressure generator fitted with a high-efficiency particulate air (HEPA) filter. This system autonomously regulated the ventilation boosters (up to four levels) to maintain a consistent pressure differential of approximately −10 to −20 Pa, monitored continuously by a differential pressure sensor (Beck Sensortechnik GmbH, Steinenbronn, Germany). The previously validated automatic operating conditions were employed in this study without modification.

### 2.3. Computational Fluid Dynamics Simulation

In this study, advanced Eulerian–Lagrangian modeling techniques inspired by recent studies were integrated to enhance the accuracy and realism of the CFD simulations [13,14]. Specifically, droplet evaporation was simulated under realistic, spatially varying humidity conditions to accurately predict the aerosol dispersion and deposition. Additionally, transient boundary conditions incorporating realistic coughing behaviors, including human head motion [22], were adopted. The detailed turbulence modeling was guided by recent direct numerical simulation (DNS) results, significantly enhancing the reliability.

The numerical simulations were conducted using ANSYS CFX software (version 2023 R1, Ansys Inc., Canonsburg, PA, USA). The shear stress transport (SST) k–ω turbulence model was employed for all the CFD simulations due to its superior accuracy in capturing complex turbulent flow characteristics, particularly for near-wall regions and under adverse pressure gradient conditions, consistent with those encountered in negative pressure containment hood scenarios. The governing equations of the SST k–ω turbulence model are described as follows:(1)$$\frac{\partial \rho }{\partial t}+\frac{\partial }{\partial x_{j}}\left(\rho U_{j}\right)=0$$(2)$$\frac{\partial \rho U_{i}}{\partial t}+\frac{\partial }{\partial x_{j}}\left(\rho U_{i}U_{j}\right)=-\frac{\partial p}{\partial x_{i}}+\frac{\partial }{\partial x_{j}}\left(\tau_{ij}-\rho \bar{u_{i}u_{j}}\right)+S_{P}$$(3)$$\frac{\partial \left(\rho k\right)}{\partial t}+\frac{\partial }{\partial x_{j}}\left(\rho U_{j}k\right)=\frac{\partial }{\partial x_{j}}\left[\left(\mu +\frac{\mu_{t}}{\sigma_{k}}\right)\frac{\partial k}{\partial x_{j}}\right]+{\;P}_{k}-{\;\beta }^{\prime }\rho k\omega +P_{kb}$$(4)$$\frac{\partial \left(\rho \omega \right)}{\partial t}+\frac{\partial }{\partial x_{j}}\left(\rho U_{j}\omega \right)=\frac{\partial }{\partial x_{j}}\left[\left(\mu +\frac{\mu_{t}}{\sigma_{\omega }}\right)\frac{\partial \omega }{\partial x_{j}}\right]+\alpha \frac{\omega }{k}P_{k}-\beta \rho \omega^{2}+P_{\omega b}$$

In addition to the independent variables, the density (*ρ*) and the velocity vector (*U*) are treated as known quantities from the Navier–Stokes method. *P_k_* is the production rate of turbulence, which is calculated as noted in the k–ε model, *τ* is the molecular stress tensor, and *S_P_* is the source term of the momentum by particle.

The unknown Reynolds stress tensor, $\rho \bar{u_{i}u_{j}}$, is calculated as follows:(5)$$-\rho \bar{u_{i}u_{j}}=\mu_{t}\left(\frac{\partial U_{i}}{\partial x_{j}}+\frac{\partial U_{j}}{\partial x_{i}}\right)-\frac{2}{3}\delta_{ij}\left(\rho k+\mu_{t}\frac{\partial U_{k}}{\partial x_{k}}\right)$$

The computational domain was discretized into approximately 6,689,889 elements, each with a uniform element size of 5 mm. This mesh size was chosen based on a rigorous grid independence test conducted previously, confirming the minimal solution variation (less than 1.26%) and stable numerical accuracy at this resolution.

Boundary conditions were clearly defined for realistic aerosol dispersion and airflow behavior. The inlet boundary condition at the patient’s mouth simulated cough-induced airflow using a previously validated cough velocity profile and aerosol particle size distribution [22,23]. The outlet boundary condition, defined at the suction port, applied a relative pressure of −10 Pa to replicate the negative pressure conditions. Open regions, excluding the patient and chamber surfaces, were assigned opening boundaries with a relative pressure of 0 Pa to represent ambient atmospheric conditions. All the patient and chamber surfaces were modeled with no-slip wall boundary conditions to accurately capture the airflow and particle deposition [12,13,14]. A schematic illustration clearly representing the boundary condition locations (outlet boundary in red, opening boundaries in blue, and wall boundaries in gray) is provided to enhance the methodological transparency (Figure 3).

The CFD methodology maintained consistency with previously established validated modeling parameters. Although the details of the droplet introduction were not clearly delineated in the initial version of this manuscript, droplets were explicitly modeled and introduced into the computational domain following previously established protocols. Aerosol droplets were injected using an inlet boundary condition, specifically representing the velocity profile of a realistic cough event based on prior validated experimental data [19]. The droplet size distribution employed was described by the Rosin–Rammler distribution, reflecting empirical data previously reported in the literature [18]. Each introduced droplet underwent subsequent breakup processes, which were rigorously modeled using the Taylor analogy breakup (TAB) model. The TAB model describes the droplet distortion using a time-dependent deformation equation, incorporating aerodynamic forces, surface tension, and viscous effects. The breakup dynamics are governed by the following equation:(6)$$y\left(t\right)={We}_{C}+e^{-\frac{t}{t_{D}}}\left[-{We}_{C}\cos\omega t-\frac{{We}_{C}}{\omega t_{D}}\sin\omega t\right]$$

Here, the characteristic breakup time $t_{D}$ and oscillation frequency ω are defined as follows:(7)$$t_{D}=\frac{2\rho_{P}r^{2}}{C_{d}\mu_{P}},\;\omega^{2}=\frac{C_{k}\sigma }{\rho_{P}r^{3}}-\frac{1}{t_{d}^{2}}$$

The dimensionless critical Weber number $We_{C}$ is expressed as follows:(8)$$We_{C}=We\frac{C_{f}}{C_{k}C_{b}}$$

The empirical constants for these parameters were carefully chosen based on the validated literature values: critical amplitude coefficient (*C_b_*) = 0.5, damping coefficient (*C_d_*) = 5.0, external force coefficient (*C_f_*) = 1/3, restoring force coefficient (*C_k_*) = 8.0, new droplet velocity factor (*C_ν_*) = 1.0, and energy ratio factor (*K*) = 10/3.

Furthermore, aerosol droplets within the domain were subjected to multiple physical forces, including aerodynamic drag, buoyancy, rotation-induced forces (Coriolis and centripetal), virtual mass effects, and fluid pressure gradients. These forces were explicitly accounted for using established models. Specifically, the aerodynamic drag force was calculated with a drag coefficient ($C_{D}$) of 0.44 using the following formula:(9)$$F_{D}=\frac{1}{2}C_{D}\rho_{F}\frac{\pi d_{P}^{2}}{4}\left|U_{F}-U_{P}\right|\left(U_{F}-U_{P}\right)$$

Other significant forces, such as buoyancy, virtual mass, rotation-induced forces, and fluid pressure gradients, were computed using standardized equations, thereby ensuring comprehensive and realistic modeling of the droplet–fluid interactions. A two-way coupling approach was adopted that integrated a particle momentum source term into the fluid momentum equation to capture the dynamic interactions between the droplets and the airflow realistically. This momentum source was continuously updated based on the droplet locations and velocities at each computational timestep.

Cough-induced airflow conditions within the chamber were simulated using the cough velocity profile from earlier studies [23]. The aerosol particle behavior, including droplet breakup, was modeled using the Rosin–Rammler distribution combined with the Taylor analogy breakup (TAB) model. The particle trajectories considered the drag, buoyancy, virtual mass, pressure gradient forces, and domain rotation effects, such as the Coriolis and centripetal forces. A two-way coupling approach was utilized, enabling interactive dynamics between the aerosol particle trajectories and the airflow patterns.

Two structural modifications to the previously validated negative pressure chamber design (detachable aerodynamic cap and air curtain system) were systematically evaluated using CFD simulations and compared against the baseline design to assess their relative efficacy in terms of aerosol containment. The CFD methodologies implemented demonstrated excellent predictive accuracy (average 81.27–95.35%) compared to the experimental measurements, confirming the reliability and robustness of the presented results.

In a previous publication, the CFD was comprehensively validated by dividing the open area of the negative pressure hood into nine distinct regions and comparing the CFD predictions of the aerosol droplet leakage mass flows with the corresponding experimental sensor measurements. This comparison clearly demonstrated strong consistency between the simulated and experimentally measured aerosol distributions. For instance, the CFD results showed leakage predominantly occurring at the opposite side of the coughing direction and at the lateral regions, accurately replicating the experimental measurements. Therefore, this study utilized this rigorously validated CFD methodology to confidently explore various design modifications prior to their experimental implementation.

Although explicit modeling of vapor diffusion as a separate scalar transport process was not employed in this study, the vapor dispersion was implicitly captured through detailed aerosol droplet evaporation modeling within the Eulerian–Lagrangian CFD framework. Specifically, the aerosol droplets in the simulation evaporated due to spatial humidity variations, temperature gradients, and local airflow velocity differences, reflecting realistic clinical environments. The droplet evaporation rates and aerosol size changes were carefully modeled according to previously validated empirical data and experimentally established cough aerosol size distributions [22,23]. Thus, while separate vapor diffusion equations were not explicitly solved, the mass transfer due to evaporation inherently accounted for the essential vapor dispersion phenomena, ensuring realistic representation of the aerosol behaviors and dispersion patterns.

### 2.4. Polyalphaolefin Particle Leakage Test

In this study, the aerosol particle leakage from a negative pressure chamber was assessed using the polyalphaolefin (PAO) particle method. This method is commonly used to test the integrity of HEPA filters, following the ISO 14644-3 standard [24]. Inside this chamber, aerosolized PAO particles were generated at concentrations of between 100 and 120 µg/L, significantly higher than the ISO standard recommendation (20–30 µg/L). This approach was adopted to rigorously evaluate the chamber performance under severe conditions. An aerosol photometer (Model PH-5, Tec Services, Inc., Atlanta, GA, USA) was used to measure the particle leakage. Significant leakage was defined as any amount exceeding 0.3%, and all leakages above 0.1% were recorded.

The open side of the chamber was divided into nine measurement sections (Figure 4C [20]) to evaluate the spatial leakage patterns. The particle leakage was measured every 10 s for 90 s at a distance of 15 cm from the chamber’s opening. Aerosols were dispersed within the chamber in three distinct directions: toward the suction hole, opposite to the suction hole, and directly toward the center of the ceiling. Each test was repeated five times (Figure 4B).

Furthermore, simulations of the aerosol dispersion were conducted under conditions resembling coughing and normal breathing, following methods identical to those reported in previous studies [20]. The aerosol leakage performance of the negative pressure chamber was systematically compared under two conditions: with and without the aerodynamic cap structure. In the coughing scenario, the PAO particle concentrations were rapidly elevated to 100–120 µg/L and released in six distinct bursts over 90 s to replicate coughing episodes. The normal breathing simulation involved a steady release of the same total amount of aerosol over 90 s using a controlled valve system. These simulations generated aerosol concentrations significantly higher than under typical physiological conditions, providing stringent test conditions for evaluating the chamber’s efficacy under both steady-state and transient aerosol release scenarios.

### 2.5. Biological Aerosol Testing

Biological validation of the aerosol containment performance of the negative pressure chamber was conducted using aerosolized bacteria at a specialized full-scale emergency department simulation facility located in Paju, Korea. This facility was designed to replicate the layout and environmental conditions of the Seoul National University Hospital Emergency Department. *Bacillus subtilis* (ATCC 6051) was used because it is a well-established, non-pathogenic bioaerosol surrogate that is commonly used in containment testing. The bacterial cultures were prepared following standardized procedures to achieve a known initial concentration suitable for aerosolization.

Bacterial aerosol particles were generated using a collision nebulizer (Model NSF CN-311), externally supplied with compressed air from outside the containment area to minimize the potential airflow interference within the chamber. The aerosolized bacteria were continuously dispersed at a uniform rate for 60 min. After completing the aerosol dispersion, an additional stabilization period of 30 min was allowed. Airborne bacteria were then actively sampled over 20 min using a calibrated microbial air sampler (Model B-301, BNF Korea, Incheon, Republic of Korea). The settling plates were continuously exposed from the start of the aerosol dispersion until the completion of the airborne sampling (110 min total). At the end of this period, all the settling plates were simultaneously collected by the experimenters, who were wearing sterile gowns to avoid contamination.

The bacterial contamination was quantitatively assessed through active air sampling and passive settling plates placed at predetermined locations inside and around the negative pressure chamber, as shown in Figure 5. Passive bacterial sampling was performed by exposing 90 mm diameter Tryptic Soy Agar (TSA) plates (90 mm diameter, gamma irradiated; SAMWOO S&T Co., Ltd., Incheon, Republic of Korea) at locations marked with red circles, as shown in Figure 5. Active air sampling was conducted using the same type of TSA plates at three locations indicated by blue triangles in Figure 5, utilizing a calibrated microbial air sampler.

The number of colonies in the dishes was directly quantified when clearly countable. In cases where the colonies were too numerous to count (indistinct or overlapping colonies), the entire sample was collected and serially diluted up to 10^10^ fold. Subsequently, 10 µL aliquots from each dilution were plated separately and incubated. Then, the colonies were enumerated to accurately quantify the colony-forming units (CFUs). Recognizing that the colony numbers could have increased significantly during the 20 h incubation period, the bacterial growth rates for *Bacillus subtilis* (approximately doubling every 30 to 60 min under optimal laboratory conditions) were considered. Accordingly, the final CFU counts were adjusted to account for the estimated proliferation during incubation.

The control condition was established without using the negative pressure hood and followed the previously standardized aerosolization conditions routinely set for other experiments conducted in the same simulation facility. Under these control conditions, colonies formed on both the settling plates and active air sampling plates grew to an uncountable density. Consequently, serial dilutions up to 10^10^ fold were performed to accurately quantify the colony-forming units (CFUs) per plate. Even after performing 10^10^ fold dilutions, the settling plates exhibited too many colonies to count, indicating an extremely high initial aerosolized bacterial concentration. Considering that *Bacillus subtilis* has a doubling time of approximately 30 to 60 min under optimal laboratory conditions, the number of colonies likely increased by a factor exceeding one million (approximately 2^20^) during the 20 h incubation period. Thus, it can be conservatively inferred that the original number of aerosolized CFUs was at least 10^4^ or higher.

For precise quantitative comparisons, ideally, controls should be established with approximately 100–1000 CFUs. However, this study intentionally created an extreme aerosolization environment producing at least 10^10^ CFUs on the settling plates to evaluate the containment performance capable of achieving a bacterial leakage reduction of up to one millionth (10^6^) of the original concentration.

In the case of the active air sampling plates, the colonies were clearly countable after 10^7^-fold dilution, typically showing only 1 or 2 colonies. Given that the experimental objective was to achieve less than 1% bacterial aerosol leakage from the hood, the occurrence of countable colonies on each plate indicates that the bacterial leakage was effectively controlled to below 1/100 (1%) of the initially aerosolized bacterial population.

Biological aerosol testing was conducted to rigorously confirm and complement the aerosol containment findings obtained from the CFD simulations and PAO particle leakage testing, thereby ensuring comprehensive validation of the practical containment capabilities of the negative pressure chamber.

## 3. Results

### 3.1. CFD Results

Based on the previously validated CFD methodology and its proven predictive accuracy, the CFD simulations in this study served as a reliable basis for evaluating the structural modifications aimed at improving the aerosol containment performance.

Figure 6 presents the total particle mass flow rates at the mouth (injection), outlet (extraction), and opening for the baseline case (no cap) over the first 33.2 s. The opening mass flow rate peaks sharply, subsequently decaying quasi-exponentially as inward flow is re-established by the negative pressure. The peak outward mass flow rate is 2.8 × 10^−6^ kg/s and occurs at t = 0.23 s. At t = 0.35 s, approximately 90% of the total escaped mass has already left the enclosure. At t ≈ 0.60 s, the figure reaches 99%. Integration of $\dot{m}\left(t\right)$ over the first 33.2 s yields a cumulative escaped mass of 4.30 × 10^−6^ kg, corresponding to 72.66% of the injected particle mass. Although the leakage rapidly diminishes, the magnitude of the initial burst significantly exceeds the recommended exposure thresholds for highly infectious aerosols in confined clinical environments.

These results illustrate inherent limitations in the conventional CFD approaches, notably their inadequate representation of the complex turbulent airflow–aerosol interactions and insufficient modeling of the droplet evaporation processes under realistic humidity conditions [12,14]. Furthermore, accurately capturing transient respiratory behaviors, including realistic coughing actions and human head motions, remains essential yet challenging, necessitating advanced modeling strategies to enhance reliability in aerosol containment predictions [13]. This study addressed these critical limitations by integrating sophisticated turbulence modeling, realistic transient boundary conditions, and detailed environmental parameters into our CFD methodology, significantly improving the prediction accuracy and robustness.

In direct response to the aforementioned CFD limitations, a quarter-circular aerodynamic cap was attached to the enclosure (Figure 7B). Its convex outer surface effectively deflects the cough jet into the extraction stream, and the smooth inner lip stabilizes the shear layer at the rim. A coherent inward flow develops across the opening, limiting the peak outward velocity to 0.139 m/s. Throughout the 33.2 s simulation, no particles escape through the opening, giving an escape fraction below the solver’s numerical detection limit (<0.01%).

Maps of the turbulent kinetic energy 5 mm inside the opening show a uniform, low-intensity field with a mean value of 3.1 × 10^−3^ m^2^/s^2^. The enclosure purges rapidly. Specifically, 95% of injected particles exit via the suction port within 1.5 s, and the particle population is reduced to less than one part per million after 3.7 s.

To enable direct comparison with the baseline, Figure 7C shows a plot of the time history of the total particle mass flow rates at the mouth (injection), outlet (extraction) and opening. The graph shows a rapid increase in the extraction flow that surpasses the injection within 0.04 s, followed by a monotonic decline as the chamber clears. Here, 50% of the injected mass leaves through the outlet in 5.7 s, 80% in 10.7 s, and 95% in 15.4 s. The opening curve remains close to zero throughout, confirming complete containment without any outward spike comparable to the baseline surge.

Three additional simulations were performed to evaluate how well the cap maintains containment when the suction hose is moved in ways that commonly occur during procedures. In the first, bilateral scenario, the patient’s head is rotated 90° to the left, and identical hoses draw simultaneously from the right and left flanks at mouth height. Figure 8A confirms that the opening mass flow rate remains effectively at zero, indicating negligible cumulative escape. Half of the injected mass leaves in 1.8 s, 95% in 4.7 s, and the remaining concentration drops below the detection limit by 7 s—within 0.3 s of the centered suction benchmark. A companion slice taken 5 mm above the enclosure floor near the outlet (Figure 9) shows opposing entrainment plumes merging into a broad, symmetric vortex pair that sweeps the cough jet downward and then radially inward across the bottom surface toward the cap throat, preventing outward recirculation.

The second run imitates a front-to-rear switching routine. Specifically, a frontal hose captures the jet until t = 0.20 s, after which suction is transferred to a port 250 mm behind the patient’s head. As the hose changes position, the opening signal increases only slightly above the baseline noise band (Figure 8A). Over the whole 33.2 s simulation, the outward leakage remains below the numerical detection threshold (<0.01% of the injected mass), and 95% of particles are cleared in 6.2 s. This is about 0.7 s slower than that noted with a fixed frontal hose.

The final rear-to-front sequence is intentionally the most demanding. Suction begins behind the head and moves to the front once coughing stops. While the rear hose is active, a wake eddy forms. When the hose shifts forward, the eddy detaches and nudges a small parcel of particles toward the opening. The opening trace peaks at 5.1 × 10^−8^ kg/s (Figure 8C), amounting to only 0.05% of the injected mass. Despite this transient, 95% of particles are expelled within 2.0 s.

Across all three maneuvers, the opening mass flow curve remains close to zero and never approaches the surge recorded for the uncapped baseline. The aerodynamic cap therefore preserves the practical containment and rapid turnover whether suction is applied bilaterally, shifted from front to rear, or moved from rear to front during a procedure.

The simulations demonstrate that a simple aerodynamic cap can transform a marginal baseline enclosure into a robust containment system. The cap reduces the worst-case leak from 72.66% of the injected mass to 0.08%, halves the time to clear 50% the particles, and maintains complete practical containment even when the suction hoses are moved or operated bilaterally. The wall shear stresses at the patient surfaces remain far below the comfort thresholds, so the design adds no perceptible physiological burden. Because the cap can be manufactured by injection molding or three-dimensional printing and retrofitted to existing devices, it offers an immediate, low-cost upgrade path for negative pressure enclosures used during aerosol-generating medical procedures.

### 3.2. Polyalphaolefin Particle Leakage Tests

Using the PAO particle leakage method, the aerosol containment performance of a negative pressure chamber was evaluated under breathing and coughing simulation conditions, with and without the newly introduced cap structure. Measurements were systematically obtained at nine predefined positions around the chamber opening, with aerosol dispersion conducted in three distinct directions: ceiling directed, toward the suction hole, and opposite the suction hole. Each directional condition was measured separately at all nine positions (Figure 10).

Consistent with our previous study [20], we discovered that the average aerosol leakage in the configuration without the cap did not exceed the critical threshold of 0.3% (Figure 10A,C). However, transient leakage peaks surpassing this threshold were frequently observed (Figure 10C), particularly during simulated coughing directed away from the suction hole. For instance, the highest instantaneous leakage without the cap reached approximately 0.657% at position 5 during the initial 10 s interval of coughing in the opposite direction to the suction hole, significantly exceeding the acceptable containment limit (Figure 10C, Area 5, orange line). Introducing an aerodynamic cap considerably improved the containment performance, substantially reducing the instantaneous aerosol leakage. With the cap structure in place, the aerosol leakage consistently remained well below the critical threshold of 0.3% across all the measurement positions during the breathing and coughing simulations (Figure 10B,D). Even during the coughing simulations, the maximum aerosol leakage was substantially reduced to approximately 0.086% at position 5 (Figure 10B, Area 5, blue line), underscoring the cap’s significant containment improvement compared with the configuration without a cap. During simulated breathing, the aerosol leakage was negligible at all the measurement locations, further emphasizing the overall effectiveness of the cap structure in maintaining containment integrity. Figure 11 comparatively indicates the maximum aerosol leakage of the four conditions (breathing with/without cap, coughing with/without cap).

Time-dependent analysis of the PAO particle leakage test results showed a clear difference between the conditions with and without the aerodynamic cap structure (Table 1, Appendix A and Figure 12). In the absence of the cap, particularly under the coughing simulation, we observed significant transient leakage within the initial 10–20 s, predominantly at the ceiling and on the opposite side relative to the suction hole. The highest mean leakage rate occurred at the ceiling, reaching approximately 0.31% and exceeding the predefined critical threshold of 0.3%. Additionally, the maximum mean leakage rate on the opposite side reached approximately 0.12%. The leakage during breathing without the cap was generally lower, and transient leakage peaks averaging approximately 0.05% were still observed at the ceiling during the initial 10 s interval.

In contrast, the cap structure substantially reduced the aerosol leakage under breathing and coughing conditions. Notably, under coughing conditions, the average leakage rate consistently remained below 0.1% across all the time intervals and measurement points. The maximum leakage was observed at the ceiling (approximately 0.02%), well below the critical threshold (Table 1). Under breathing conditions, the leakage was negligible across all the intervals and directions and remained consistently below 0.04%.

Moreover, the standard deviation (SD) of the leakage rates significantly decreased with the introduction of the cap structure. This reflects improved stability and reproducibility in the containment performance compared with the no-cap scenario. These results show that the aerodynamic cap effectively guided the internal airflow toward the suction hole and efficiently suppressed the transient aerosol leakage peaks. Consequently, our results strongly show the proposed cap structure to be a significant improvement. Figure 12 shows the comparative results of the mean leakage and the standard deviation of the six conditions (with/without cap, ceiling, opposite, same direction of aerosol disperse).

### 3.3. Biological Leakage Test Results

We further validated the aerosol containment efficacy of the negative pressure chamber equipped with an aerodynamic cap structure using biological leakage testing. Aerosolized *B. subtilis* (ATCC 6051), a robust bacterial surrogate commonly employed in bioaerosol evaluation, was used in these experiments.

Two sampling methodologies were used in the quantitative assessments of the bacterial leakage: passive settling plates to detect the surface deposition of bacteria at predefined locations (C1–C11) and active air sampling at five aerosol sampling positions (positions 1–3) inside and immediately around the chamber (Figure 5).

Under control conditions without the chamber, the bacterial contamination on the settling plates consistently exceeded the quantifiable limits (over 10^10^ CFU/plate), demonstrating substantial bacterial aerosolization. Similarly, the air sampling plate bacteria concentration in the control scenario varied widely, exceeding (10.8 × 10^6^ CFU/plate) and often surpassing the detection limit.

In contrast, use of the negative pressure chamber and aerodynamic cap structure dramatically reduced the bacterial leakage. Among the settled plates, the highest recorded bacterial colony counts were 86 CFU/plate at location C1 and 11 CFU/plate at location C2, while the counts at other sampling positions were lower than 10 CFUs. Active air sampling showed minimal bacterial escape, with a maximum recovered airborne bacterial count of only two CFUs at aerosol sampling position 3. Most of the other sampling locations produced values between 0 and 2 CFUs (Table 2).

These results clearly showed that the chamber and cap structures effectively achieved the intended aerosol containment goal, exceeding the predefined target of at least 99% bacterial containment. Given the extraordinarily high bacterial concentration in the control conditions (over 10^10^ CFU/mL), the observed containment performance (below 100 CFUs at the highest measurement point) shows a reduction that significantly surpasses the 99% containment threshold. This outcome fully meets and exceeds the initial containment objectives.

## 4. Discussion

The findings from this study demonstrated the enhanced aerosol containment performance achieved by incorporating an aerodynamic cap structure into a negative pressure chamber. Using computational fluid dynamics (CFD), physical leakage tests, and biological validation with aerosolized *B. subtilis*, the results consistently showed superior aerosol control owing to this structural modification. The containment effectiveness consistently exceeded the target of 99%, highlighting its clinical utility in mitigating airborne pathogen transmission risks.

In conclusion, our research demonstrates that integrating advanced CFD methodologies, which incorporate realistic transient boundary conditions, sophisticated turbulence modeling, and precise modeling of droplet evaporation dynamics under realistic humidity conditions, effectively overcomes existing challenges associated with aerosol containment in negative pressure hood environments [12,13,14]. By adopting these enhanced CFD modeling approaches, our study substantially improved the reliability of the aerosol dispersion predictions and supported the development of clinically effective and practically feasible infection control solutions.

In this study, the CFD simulations provided quantitative insights into the airflow mechanisms contributing to the cap’s containment performance. The convex outer surface of the aerodynamic cap immediately redirected the cough-generated aerosol plume toward the suction port, while the inward-curved lower edge stabilized the shear layer near the opening, substantially reducing the turbulence. Specifically, the CFD simulations indicated that the mean turbulent kinetic energy around the enclosure opening decreased dramatically to approximately 3.1 × 10^−3^ m^2^ s^−2^, approximately two orders of magnitude lower than that measured in the baseline configuration without the cap.

Additional CFD simulations reflecting clinically realistic suction port configurations further validated the robust containment performance of the aerodynamic cap. Three typical clinical suction port operational patterns were evaluated: simultaneous bilateral suction, front-to-rear suction switching, and rear-to-front suction switching. Remarkably, all three scenarios exhibited minimal aerosol leakage (≤0.05% of the injected mass). Even in the most challenging scenario—switching suction from the rear to the front—a transient vortex briefly directed aerosols toward the opening. However, the resulting peak outward flux remained very low (5.1 × 10^−8^ kg·s^−1^), yielding a cumulative aerosol leakage of only 0.05%. These results underscore the aerodynamic cap’s ability to reliably maintain effective containment and rapid clearance under diverse suction port configurations representative of actual clinical practice.

Furthermore, the planar flow field visualization near the enclosure floor clearly demonstrated the flow patterns responsible for the aerodynamic cap’s superior containment. Specifically, the bilateral suction configuration generated two symmetrical vortices near the enclosure bottom, creating strong downward and inward flow fields that effectively entrained the aerosol particles toward the outlet and prevented recirculation toward the opening. This detailed CFD-based flow analysis provides a clear mechanistic explanation of how the aerodynamic cap ensures robust aerosol containment in clinical settings.

A key advantage of the aerodynamic cap is its effective prevention of aerosol leakage, a common issue in conventional open-type chambers. The aerodynamic cap maintains partial openness, substantially improving the clinical practicality, unlike previous approaches that predominantly relied on a nearly complete sealing for containment [25,26]. Fully sealed systems, while effective, often restrict patient access and complicate necessary clinical interventions. Our design effectively mitigates these practical issues and achieves a comparable containment performance while preserving clinical flexibility.

Although direct CFD-to-sensor comparative data were not explicitly presented in this manuscript, rigorous CFD validation was previously conducted and clearly demonstrated in our earlier published research. In a prior study, systematic comparisons were performed by dividing the enclosure opening into nine distinct regions, carefully assessing the aerosol leakage patterns using both CFD simulations and sensor-based experimental measurements. The results consistently exhibited close agreement between the simulated aerosol distributions and the experimental observations, particularly highlighting the aerosol leakage on predominantly the opposite side of the coughing direction and in lateral regions. This robust validation provided a solid methodological foundation, enabling confident reliance on the CFD predictions for structural design improvements in the current study [20].

The aerodynamic cap structure introduced in this study was specifically designed to address the aerosol leakage predominantly observed in the upper region of the enclosure opening during coughing events, as demonstrated in previous research. Earlier studies indicated that significant aerosol leakage occurs primarily due to strong turbulent jet flows and vortices generated by coughing, resulting in high aerosol concentrations escaping through the upper portion of the enclosure. The aerodynamic cap was thus introduced as an innovative measure aimed explicitly at redirecting the internal airflow inward and stabilizing these turbulent vortex structures, effectively minimizing the aerosol leakage in this critical region. Given the novel nature of this cap design, extensive parametric optimization and detailed geometric analyses were not conducted at this stage; rather, the present investigation focused on empirically demonstrating its fundamental feasibility and initial effectiveness in controlling aerosol dispersion. Comprehensive geometric optimization and further theoretical analyses of the cap’s specific dimensions and configurations are intended for future research.

The negative pressure switching cycles used in this study were carefully selected based on established clinical guidelines and recent engineering insights. The Centers for Disease Control and Prevention (CDC) recommends that negative pressure isolation systems maintain at least 2.5 Pa of negative pressure, accompanied by airflow velocities typically ranging from 0.2 to 0.5 m/s, to effectively manage aerosol containment. Additionally, recent investigations using particle image velocimetry (PIV) techniques have highlighted that periodic switching or directional adjustments in negative pressure airflow significantly improve the aerosol removal efficiency, prevent particle accumulation, and minimize the stagnation zones within isolation chambers [21]. Informed by these standards and insights, the negative pressure directional switching parameters in this study were systematically selected to ensure effective aerosol removal, continuous airflow stability, and enhanced containment performance.

Regarding the necessity of comparative experimental verification, although direct CFD-to-sensor comparative data were not explicitly presented in this paper, rigorous CFD validation was comprehensively conducted and clearly demonstrated in a previously published study. In that prior validation, systematic comparisons were performed by segmenting the chamber opening into nine distinct regions, carefully evaluating the aerosol leakage patterns using CFD simulations and sensor-based experimental measurements. The results from that study indicated consistently high agreement between the simulated and experimentally measured aerosol distributions, highlighting the aerosol leakage predominantly occurring opposite to the coughing direction and in lateral regions. Based on this robust prior validation, the current paper utilized CFD simulation results as a reliable basis for assessing the effectiveness of the structural modifications. Additionally, all the comparative experimental data under each condition (with and without the aerodynamic cap, as well as chamber and no-chamber conditions) have been clearly presented, thereby providing comprehensive evidence demonstrating the enhanced containment performance achieved by incorporating the aerodynamic cap.

In quantitative PAO-based aerosol leakage tests, the implementation of the aerodynamic cap demonstrated clearly stabilized airflow. Specifically, under coughing conditions, the cap effectively reduced both the mean leakage amount and its standard deviation. This finding indicates that the aerodynamic cap structure significantly stabilizes the outward airflow generated during high-pressure events, such as coughing, thereby improving the containment reliability.

Biological validation confirmed that these enhancements were effective under extreme bacterial aerosol concentrations exceeding 10^10^ CFU/plate under control conditions. The chamber with the cap structure maintained bacterial contamination levels dramatically below the control conditions. The maximum surface contamination remained consistently under 100 CFUs, while the airborne contamination was negligible, reaching only 2 CFUs at most. These findings strongly affirm the cap structure’s ability to exceed the predefined 99% containment goal.

Most importantly, as described previously in Section 1, this study initially conducted a preliminary analysis of several potential structural improvement ideas and subsequently selected the most promising model for detailed investigation and mock-up testing. This strategic approach significantly reduced the experimental costs and time compared to performing comprehensive mock-up tests for all the initial design ideas, further highlighting the practical value and efficiency of the methodology employed in this research.

Despite these promising outcomes, this study has some limitations that warrant further research. The data generated under controlled laboratory conditions must be validated in real-world clinical settings. The structural durability, long-term maintenance, and ease of operational use under diverse clinical scenarios have not been comprehensively assessed and require further investigation. The biological tests only compared the conditions with no chamber versus those using the chamber equipped with the aerodynamic cap; thus, comparative data involving the chamber without the aerodynamic cap were not obtained. Additionally, the lack of repeated experiments limits the generalizability of these findings. However, given that the test conditions were extremely rigorous, it can be reasonably anticipated that repeated experiments under similar conditions would yield comparable outcomes. Furthermore, for control groups exhibiting colony growth too dense to count directly, reliance on serial dilution and incubation steps makes it difficult to interpret the results purely as initial bacterial concentrations. Hence, establishing control conditions optimized for more accurate quantitative assessments will be essential in future studies. Moreover, our biological validation used a single surrogate bacterium; therefore, further validation using pathogenic viruses and clinically significant bacterial species is necessary. Future research incorporating stringent biosafety protocols to validate their effectiveness against various infectious agents is ongoing.

## 5. Conclusions

In this study, the aerosol containment performance of a negative pressure chamber enhanced with an aerodynamic cap structure was validated. This innovative design effectively met and surpassed the target containment efficiency of 99%, significantly reducing the aerosol leakage, especially under challenging conditions such as transient coughing events. Compared to our previous open-type chamber design [9], the aerodynamic cap substantially reduced the aerosol leakage, notably addressing the transient leakage peaks occurring opposite to the suction hole during coughing events. Previous transient leakages revealed the vulnerabilities of fully open structures to sudden airflow disturbances. By overcoming this limitation, incremental design improvements demonstrated that substantial containment efficacy is achievable without compromising the essential clinical openness. Continued structural improvements to open-type chambers are expected to broaden their clinical applications and strengthen their infection control capabilities.

The findings underscore the critical importance of employing advanced CFD modeling strategies, including with realistic transient boundary conditions, detailed turbulence modeling, and accurate humidity representations, for designing and evaluating aerosol containment systems [12,13,14]. Integrating these advanced methodologies significantly enhanced the accuracy of the aerosol dispersion predictions, thereby contributing substantially to the practical containment improvements. Ultimately, the improved aerodynamic cap design not only maintains essential clinical accessibility and practicality but also reinforces healthcare workers’ safety, supporting its widespread adoption during aerosol-generating procedures.

However, to enhance the practical utility of these results, comprehensive assessments of the structural durability, long-term maintenance, and ease of operational use under diverse clinical scenarios are required. Additionally, validation of the method’s effectiveness against actual pathogenic microorganisms is necessary for future investigations.

## Figures and Tables

**Figure 1 bioengineering-12-00624-f001:**
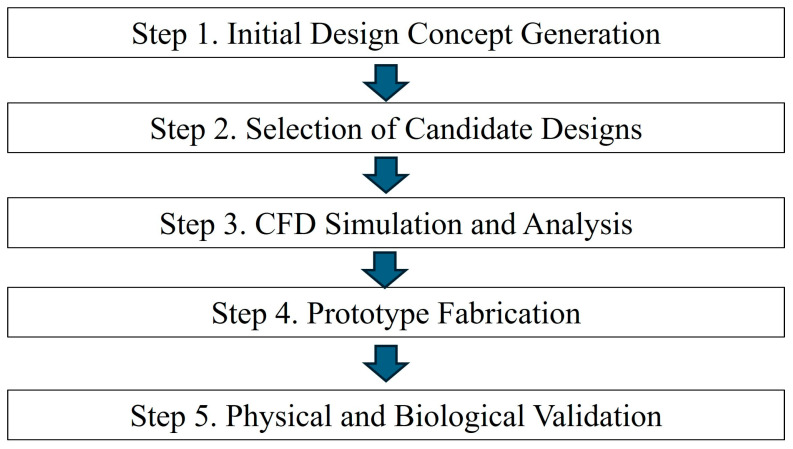
Overall validation flowchart.

**Figure 2 bioengineering-12-00624-f002:**
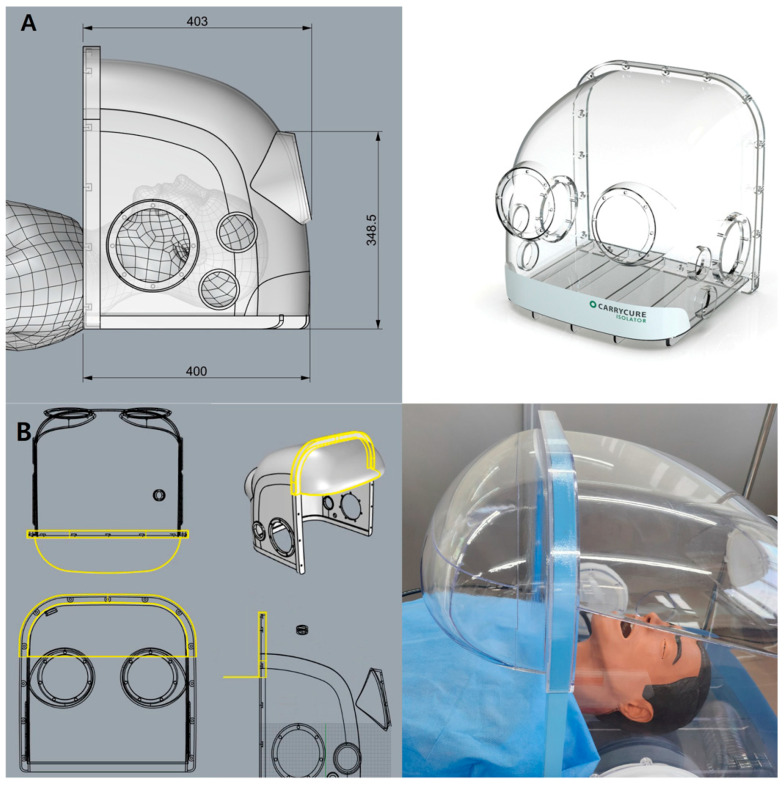
Design of the negative pressure hood (**A**) and aerodynamic cap structure (**B**). (**A**) Design of the previously validated negative pressure hood [20]. (**B**) Newly introduced detachable aerodynamic cap structure designed to improve aerosol containment. (**Left panel**) Schematic design illustrating the structural integration of the cap (yellow line). (**Right panel)** Actual image of the installed aerodynamic cap structure on the negative pressure chamber.

**Figure 3 bioengineering-12-00624-f003:**
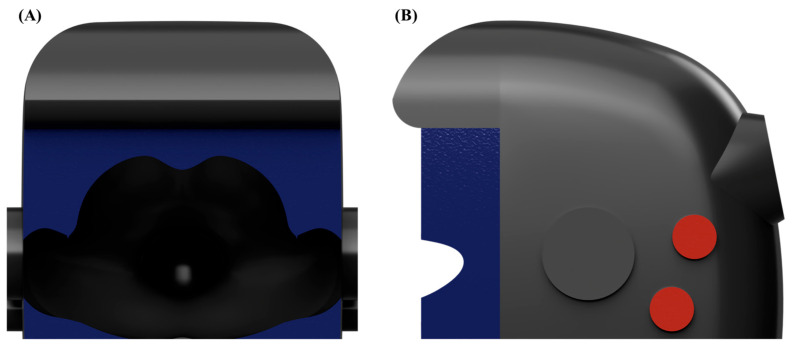
Computational domain and boundary conditions for the CFD simulation. (**A**) Front view and (**B**) side view. Blue regions indicate the opening boundary conditions (relative pressure: 0 Pa), red region indicates the outlet (suction) boundary condition (relative pressure: −10 Pa), and dark gray regions indicate the wall boundary conditions with no-slip conditions applied.

**Figure 4 bioengineering-12-00624-f004:**
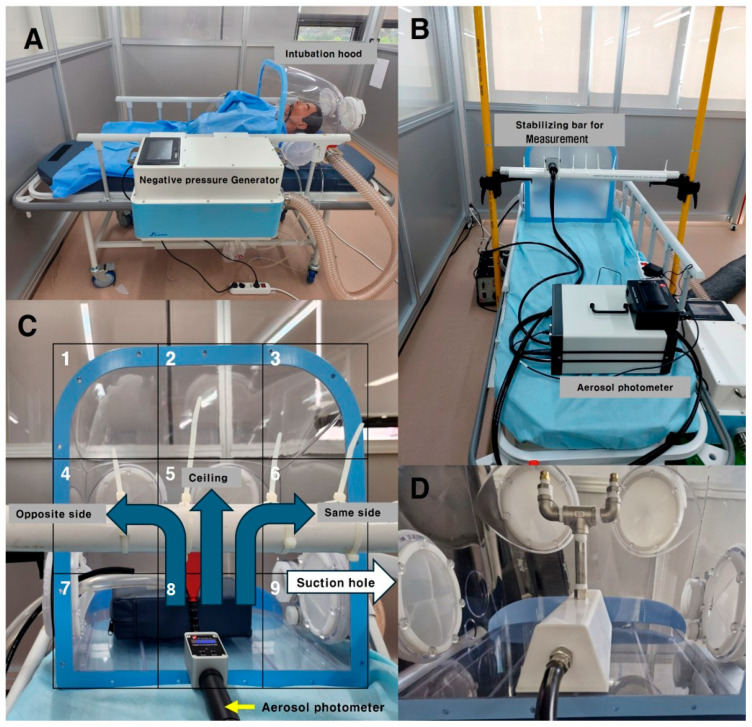
(**A**) An example of an intubation hood and a negative pressure generator with a patient. (**B**) An aerosol photometer was used to measure the particle leakage from the intubation hood; a stabilizing bar was used to ensure consistent measurement locations for the detector. (**C**) The open area of the intubation hood was divided into nine sections (indicated by numbers 1–9), and the PAO particles were dispersed in three directions (indicated by blue arrows); the suction hole was located on the inside right lateral side of the intubation hood (indicated by white arrow). (**D**) The PAO aerosol was dispersed using an aerosol generator with a valve (developed in-house by our research team) [20].

**Figure 5 bioengineering-12-00624-f005:**
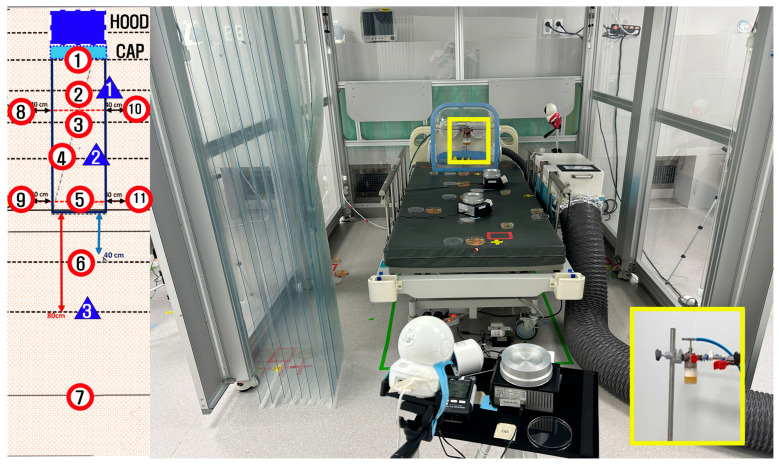
Experimental setup for biological leakage testing using aerosolized *Bacillus subtilis*. The layout diagram (**left**) illustrates the positions used for bacterial sampling within the experimental room. The negative pressure chamber is represented by the blue rectangle labeled “Hood”, with the attached aerodynamic cap indicated by the adjacent light blue rectangle (“Cap”). Settling plates (red-circled numbers, positions 1–11) were strategically placed around the chamber to collect deposited bacteria, while airborne bacteria samples (blue triangles, positions 1–3) were collected using air samplers positioned at specific distances from the chamber. The photograph (**right**) shows the actual experimental arrangement, including the chamber, aerodynamic cap, settling plates, and air samplers. The inset image (yellow box) provides a close-up view of the nebulizer device used within the hood to aerosolize bacteria during the experiments. Air discharged from the negative pressure generator passed through a high-efficiency particulate air filter; however, to exclude potential contamination from any residual leakage through the filter, the filtered exhaust was guided outside the testing area using an external hose, as depicted in the main photograph.

**Figure 6 bioengineering-12-00624-f006:**
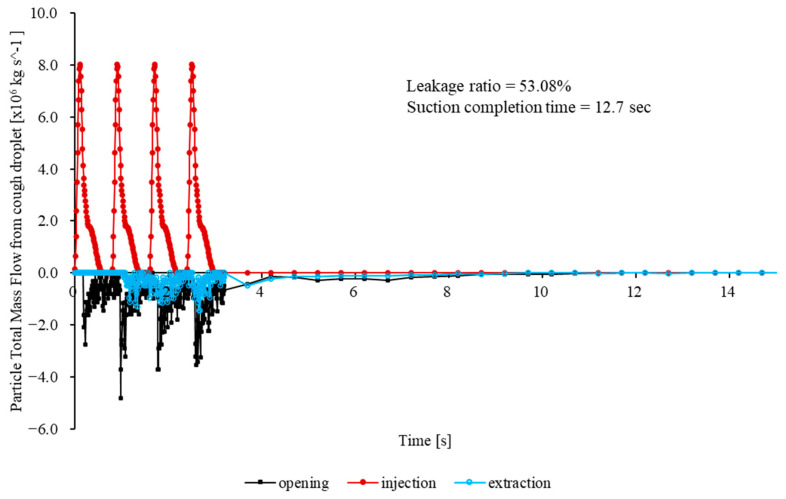
Total particle mass flow rates at the mouth (red solid circles, injection), outlet (blue open circles, extraction), and opening (black solid squares, leakage) over 33.2 s for the baseline enclosure without the cap structure.

**Figure 7 bioengineering-12-00624-f007:**
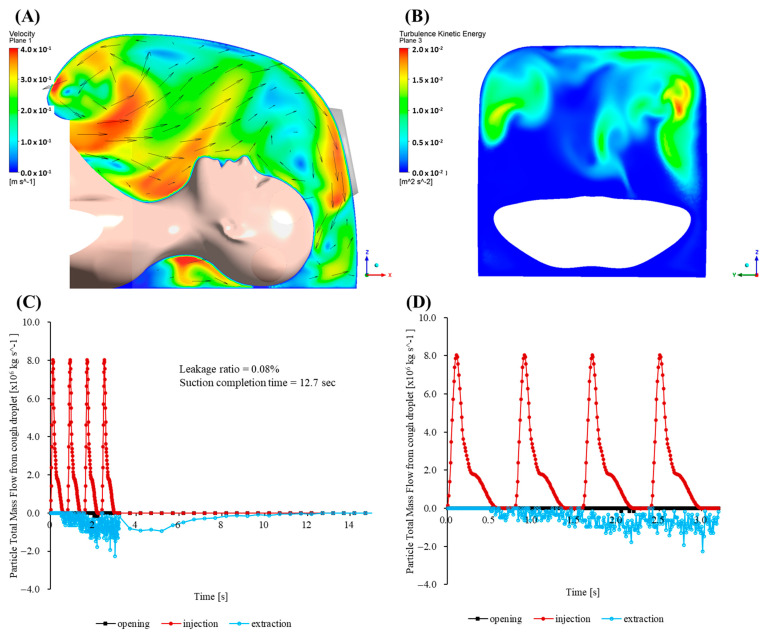
(**A**) Instantaneous velocity field for the cap-enhanced enclosure at t = 0.18 s. Colors indicate the velocity magnitude; vectors illustrate the downward deflection of the cough jet and the coherent inward flow that prevents outward leakage. (**B**) Map of the turbulent kinetic energy 5 mm inside the opening for the cap-enhanced enclosure, showing a spatially uniform low-intensity field (mean 3.1 × 10^−3^ m^2^/s^2^). (**C**) Particle total mass flow rate for the cap-enhanced enclosure: mouth injection (red solid circles), outlet extraction (blue open circles), and opening (black solid squares). (**D**) Particle total mass flow rate for the cap-enhanced enclosure (0–3 s).

**Figure 8 bioengineering-12-00624-f008:**
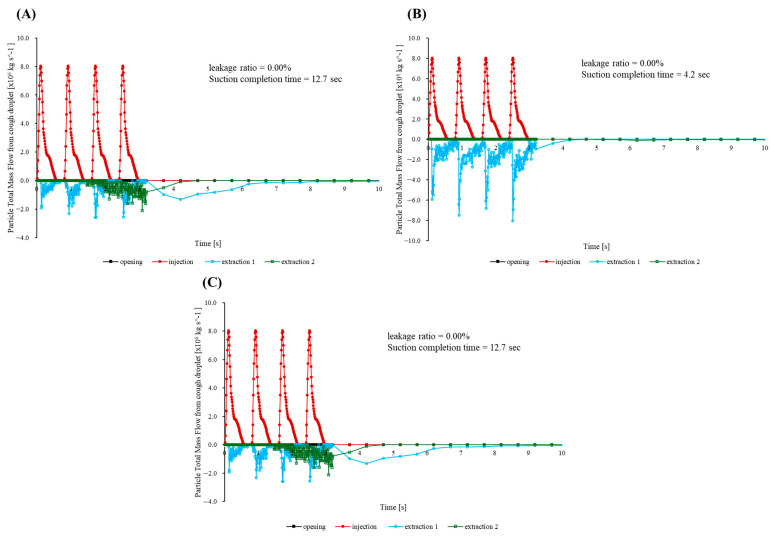
Particle total mass flow rate during (**A**) bilateral suction, (**B**) suction that switches from front to rear, and (**C**) suction that switches from rear to front: mouth injection (red solid circles), outlet extraction 1 (blue open circles), outlet extraction 2 (green open squares), and opening (black solid squares).

**Figure 9 bioengineering-12-00624-f009:**
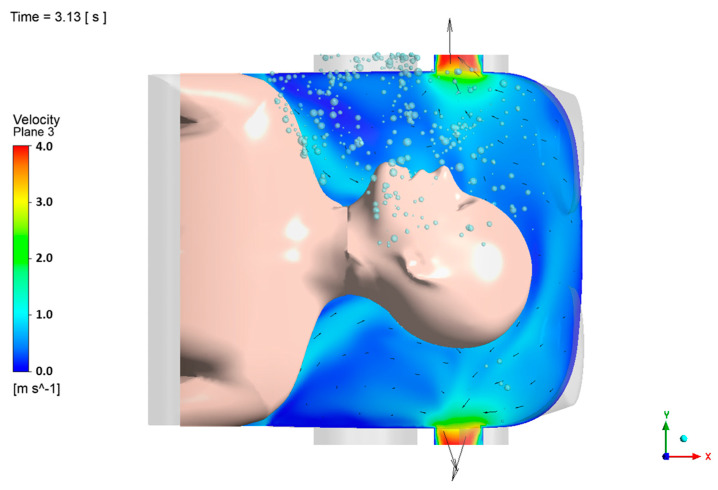
Planar velocity field 5 mm above the enclosure floor for the bilateral suction case. Colors indicate the velocity magnitude; black vectors show a symmetric vortex pair that carries the cough jet downward and radially inward toward the cap throat, preventing outward recirculation.

**Figure 10 bioengineering-12-00624-f010:**
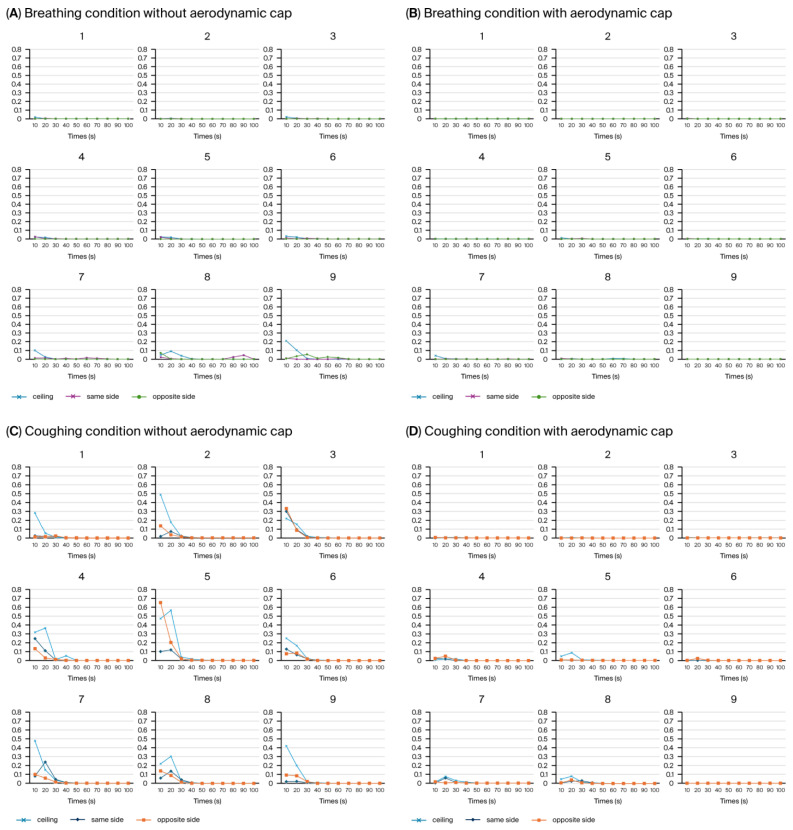
Particle leakage over 90 s, measured at nine predefined sections around the chamber opening under breathing and coughing conditions, with or without the aerodynamic cap structure. All nine graphs collectively represent the entire open area of the hood, and each number indicates the corresponding location among the nine measurement positions previously illustrated in Figure 4C. (**A**) Breathing condition without aerodynamic cap. (**B**) Breathing condition with aerodynamic cap showing aerosol dispersion toward the ceiling (blue), suction (purple), and suction side (green). (**C**) Coughing without aerodynamic cap. (**D**) Coughing with aerodynamic cap showing aerosol dispersion toward the ceiling (sky blue) [20].

**Figure 11 bioengineering-12-00624-f011:**
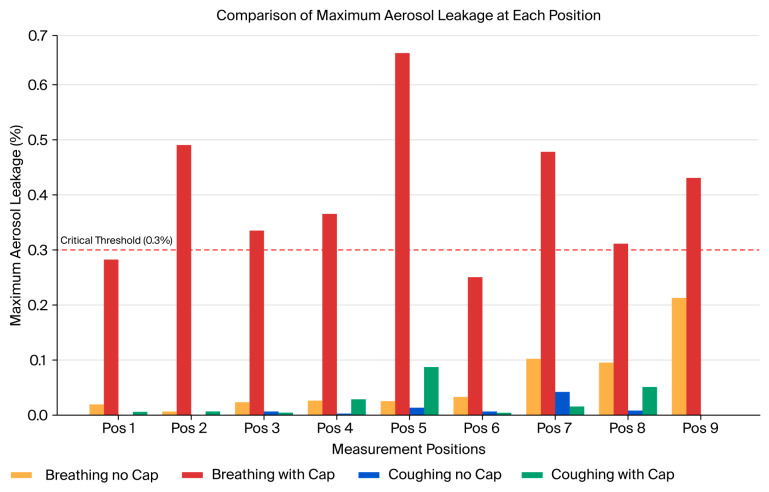
A summary and direct comparison of the maximum aerosol leakage levels recorded at each position across all the experimental conditions. Comparison of the maximum aerosol leakage measured at each of the nine designated positions within the chamber under four testing scenarios: breathing without cap, coughing without cap, breathing with cap, and coughing with cap. The red dashed line indicates the critical leakage threshold of 0.3%. Coughing without the cap notably exceeded this threshold, whereas breathing and coughing scenarios with the cap effectively maintained aerosol leakage below the critical limit, demonstrating enhanced aerosol containment.

**Figure 12 bioengineering-12-00624-f012:**
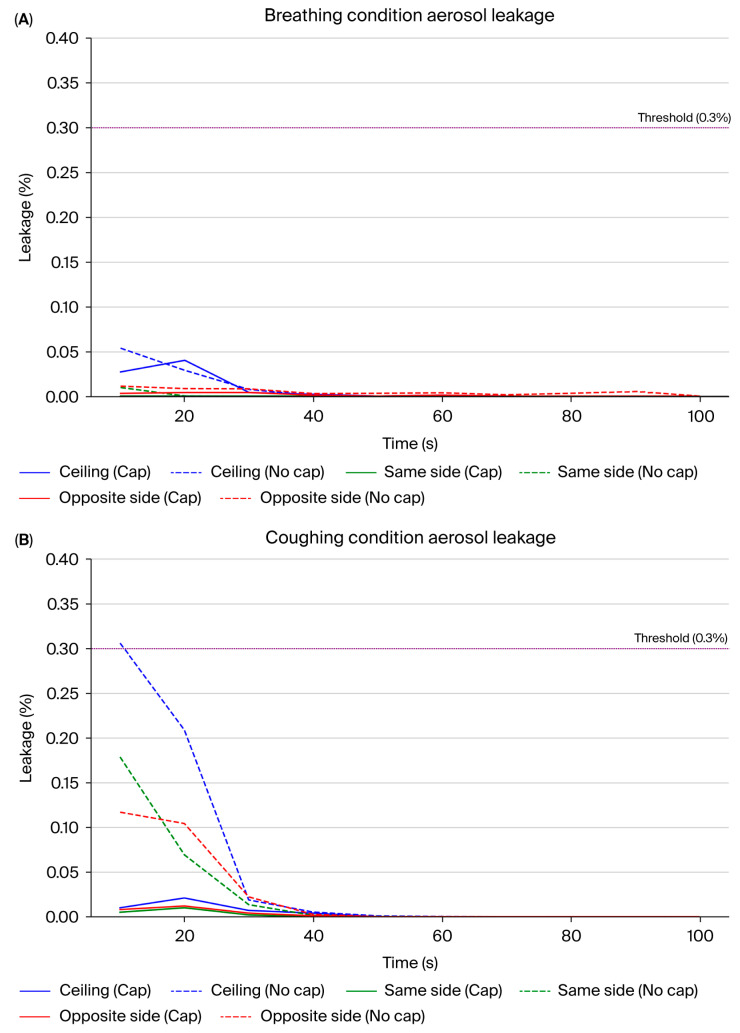
PAO particle leakage comparison with and without the aerodynamic cap structure. The particle leakage rates (mean ± standard deviation) measured over 100 s are shown under breathing (**A**) and coughing (**B**) conditions. The measurements are further divided based on the leakage directions: ceiling (blue), same side as suction (green), and the opposite side to suction (red). Solid lines represent the leakage rates with the cap structure installed, whereas dashed lines represent the leakage rates without the cap. The horizontal purple dashed line represents the predefined critical threshold of 0.3% aerosol leakage. The results clearly show the efficacy of the cap in reducing both the mean leakage rates and their variability, which is especially notable under coughing conditions, confirming the significant improvement in aerosol containment capability provided by the aerodynamic cap.

**Table 1 bioengineering-12-00624-t001:** Mean and standard deviation of droplet leakage over time.

	Breathing Condition						Coughing Condition					
Time (sec)	Ceiling		Same Side		Opposite Side		Ceiling		Same Side		Opposite Side	
	Mean	SD	Mean	SD	Mean	SD	Mean	SD	Mean	SD	Mean	SD
10	0.027	0.075	0.000	0.001	0.003	0.008	0.010	0.022	0.005	0.016	0.008	0.024
20	0.040	0.117	0.000	0.000	0.004	0.015	0.021	0.049	0.010	0.027	0.012	0.038
30	0.004	0.019	0.000	0.000	0.004	0.023	0.007	0.014	0.002	0.005	0.004	0.013
40	0.002	0.012	0.000	0.000	0.001	0.002	0.004	0.014	0.000	0.001	0.001	0.005
50	0.000	0.001	0.000	0.000	0.000	0.002	0.000	0.002	0.000	0.000	0.000	0.000
60	0.001	0.004	0.000	0.000	0.001	0.008	0.000	0.000	0.000	0.000	0.000	0.000
70	0.000	0.003	0.000	0.000	0.000	0.002	0.000	0.000	0.000	0.000	0.000	0.000
80	0.000	0.000	0.000	0.000	0.000	0.002	0.000	0.000	0.000	0.000	0.000	0.000
90	0.000	0.000	0.000	0.000	0.000	0.000	0.000	0.000	0.000	0.000	0.000	0.000
100	0.000	0.000	0.000	0.000	0.000	0.000	0.000	0.000	0.000	0.000	0.000	0.000

SD; standard deviation.

**Table 2 bioengineering-12-00624-t002:** CFU/plate of control and experiment groups.

Location/Method	Without Chamber	With Negative Pressure Chamber Featuring an Aerodynamic Cap
C1	>10^10^ CFU/plate	86 CFU/plate
C2	>10^10^ CFU/plate	11 CFU/plate
C3	>10^10^ CFU/plate	2 CFU/plate
C4	>10^10^ CFU/plate	8 CFU/plate
C5	>10^10^ CFU/plate	4 CFU/plate
C6	>10^10^ CFU/plate	7 CFU/plate
C7	>10^10^ CFU/plate	4 CFU/plate
C8	>10^10^ CFU/plate	6 CFU/plate
C9	>10^10^ CFU/plate	6 CFU/plate
C10	>10^10^ CFU/plate	8 CFU/plate
C11	>10^10^ CFU/plate	9
A1 (Triangle)	10.8 × 10^6^ CFU/plate	0
A2 (Triangle)	15.8 × 10^6^ CFU/plate	1
A3 (Triangle)	21.2 × 10^6^ CFU/plate	2

## Data Availability

No new data were created or analyzed in this study. Data are contained within the article and Appendix A.

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
