# Peer review of "Enhanced Aerosol Containment Performance of a Negative Pressure Hood with an Aerodynamic Cap Design: Multi-Method Validation Using CFD, PAO Particles, and Microbial Testing"

_bioengineering, 2025, doi:10.3390/bioengineering12060624_

Round 1
Reviewer 1 Report
Comments and Suggestions for Authors
The Authors of the entitled manuscript "Enhanced Containment of Aerosolized Particles in a Modified Negative Pressure Hood: Design Validation and Biological Performance Testing" presented an interesting article. I found the topic is relevant to the journal however, I have the following points which I would like the authors to address:
- The literature review is not sufficient to address the current models presented related to this application using CFD modelling. Please extend the literature there are many articles talks about coughing/visualization of vapor diffusion using CFD.
- The method is not clear if it was performed using k-omega or k-epsilon under CFX.
- The Mesh and computational domain are not even addressed in this paper. Also, include the boundary conditions used in this study.
- Validation: it is done but I could not see the difference between the CFD and the sensors data clearly. Authors should make that as an aim for their study.
- Explain how the droplet introduced in CFX setup.
- Have you used the vapour diffusion in this work? Explain how is implemented.
- Please avoid using "we" and write the paper using the "third person".
- Figure 8 is not clear.
- Update the conclusion section and add future work.
Reviewer 2 Report
Comments and Suggestions for Authors
Reducing human exposure to pathogenic particles is one of the essential elements of controlling the indoor environment. Preventing the spread of particles is particularly important in hospital rooms, where medical personnel work every day and patients stay, who usually have reduced resistance to infections due to illness. The most effective way to reduce the concentration of pathogenic particles is to capture them as close as possible to the source of emission. In the literature, you can find descriptions of several interesting solutions.
I consider the topic of the article to be very important, timely and specific. The article is written very well, it contains all the required parts, from a sufficient introduction to a somewhat short paragraph with conclusions.
The effectiveness of the proposed solution was verified by three well-chosen methods. I wonder whether the tracer gas method, often used in such cases, would not be one of the simpler ones. I am interested in the authors' opinion on this issue.
In general, several graphs need to be improved to make them more readable.
I also have also a few additional particular comments to the content of the presented article. The order of the comments does not reflect their significance. It results only from the order of appearance in the text of the article.
My remarks and comments:
1. Lines 207-208, “serial dilutions up to 10^10-fold” - this record should be explained exactly what diluted was. Is it 10 to the power of 10?
2. Line 230, “over the first 33.2 s” - Only 16 seconds are shown in the figure; in general, only 8 seconds would be enough.
3. Line 239, “Figure 4b” - shouldn't it be "figure 1b".
4. Figure 5 - I suggest presenting the waveforms up to 8 seconds or inserting a magnification of the waveforms covering the first 3 seconds.
5. Line 267, “bilateral-lateral” - unusual term, shouldn't it be just " bilateral".
6. Figure 8 - it's hard to see anything on these many charts
7. Figure 10 - I suggest to improve the chart, the range of the vertical axis could be 0.4%.
Reviewer 3 Report
Comments and Suggestions for Authors
This paper researches an enhanced containment of aerosolized particles in a modified negative pressure hood, give a modified negative-pressure chamber design method, the research of this paper has some significance, but there are the following problems with this paper. It is suggested to make major revisions.
(1) The term "Design Validation" is not described correctly. What kind of design does the author want to express? Please clarify the object of the design.
(2) The current abstract is not very clear. The author's specific research methods and measures cannot be seen. It is suggested that the specific new method of the research be clearly explained.
(3) In part 1, the author lacks an analysis of the latest relevant research work. It is suggested that the author supplement more specific research methods in the latest literature. And summarize the existing problems of the containment of aerosolized particles in a negative pressure hood at present.
(4) in part 2.1, the author should present an overall design process to illustrate the specific methods of the design.
(5) It is suggested that the author should present the steps of negative pressure hood design.
(6) What key parameters are mainly involved in biological performance testing? What are the testing methods for these key parameters? The author didn't state it. It is suggested that the author should have the calculation function for each parameter and the specific method for obtaining the parameters.
(7) The calculation functions in the paper have no serial numbers.
(8) It is suggested that the author describe the principle in Figure 2 in more detail.
(9) Regarding biological aerosol testing, the author did not provide the means and methods for the test. It is suggested that the author supplement.
(10) In part 3, the all figures are not very standard. Besides, there is a lack of a comparative experiment for verification.
(11) This paper belongs to the content of testing. It is suggested that the author provide sufficient theoretical models. Currently, there are no reasonable theoretical calculation models for the parameters, nor are there any reasonable processing methods.
Comments on the Quality of English Language
The expression of this paper needs further improvement
Round 2
Reviewer 1 Report
Comments and Suggestions for Authors
I would like to thank the authors for addressing all my questions and concerns. In my opinion, the revised manuscript is acceptable.
Reviewer 2 Report
Comments and Suggestions for Authors
I have read the authors' explanations regarding my comments with great care. I must admit that the authors have responded to each comment and remark in detail and in full.
The authors have modified the content of the manuscript by adding explanations and specifying statements. I believe that the content of the manuscript has been significantly improved and in its current form facilitates a better understanding of the authors' idea.
Reviewer 3 Report
Comments and Suggestions for Authors
The current revised manuscript is relatively complete. The author answers all the questions raised. Meanwhile, the author has also made supplements in the paper. I think the current revised manuscript can be published.
Comments on the Quality of English LanguageThe expression in English is reasonable.